# Thermal Reflow Simulation for PMMA Structures with Nonuniform Viscosity Profile

**DOI:** 10.3390/polym15183731

**Published:** 2023-09-11

**Authors:** Fedor Sidorov, Alexander Rogozhin

**Affiliations:** Valiev Institute of Physics and Technology, Russian Academy of Sciences, 117218 Moscow, Russia; rogozhin@ftian.ru

**Keywords:** grayscale e-beam lithography, PMMA, thermal reflow, nonuniform viscosity profile

## Abstract

This paper presents a new approach to the simulation of the thermal reflow of e-beam-exposed polymethyl methacrylate (PMMA) taking into account its nonuniform viscosity profile. This approach is based on numerical “soapfilm” modeling of the surface evolution, processed by the free software “Surface Evolver” in area normalization mode. The PMMA viscosity profile is calculated via the simulation of the exposed PMMA number average molecular weight distribution using the Monte-Carlo method and empirical formulas. The relation between the PMMA viscosity and the mobility of PMMA surface vertices was determined via the thermal reflow simulation for uniform PMMA gratings using analytical and numerical approaches in a wide viscosity range. The agreement between reflowed profiles simulated with these two approaches emphasizes the applicability of “soapfilm” modeling in the simulation of polymer thermal reflow. The inverse mobility of PMMA surface vertices appeared to be proportional to the PMMA viscosity with a high precision. The developed approach enables thermal reflow simulations for complex nonuniform structures, which allows the use of predictable reflow as a stage of 3D microfabrication.

## 1. Introduction

Thermal reflow consists in softening and material displacement that occurs in thermoplastic polymers when heated above their glass transition temperature. This process occurs due to polymer chain movement pursuing the minimization of the system total energy. The key feature of thermal reflow is the softening of sharp edges, which could be used for the modification of structures with a rectangular profile to a curved geometry or just for the reduction of surface roughness. For this reason, thermal reflow is widely used as an economical and efficient postprocessing method for applications requiring complex three-dimensional structures that are difficult to obtain with standard lithography processes. Thus, the reflow stage is applied during the fabrication of microlenses [1], antireflective moth-eye structures [2], waveguides [3], microprisms [4], microring resonators [5], etc. In many cases, thermal reflow is only used to enhance the quality of the structure surface, but more complex applications of this phenomenon also exist. For example, Kim et al. [6] proposed a method to guide the reflow along the geometric boundaries of the adjacent thermoset microstructures, Hung et al. [7] used two photoresists with different glass transition temperatures to fabricate a tilted microlens with a constrained base, and Kirchner and Schift [8] suggested a polymer reflow application for MEMS actuation.

In the light of present research, particular attention must be paid to grayscale e-beam lithography (GEBL) with a subsequent polymer reflow stage [8,9]. In this method, thermal reflow could be used for smoothing of stair-like profile obtained by e-beam exposure with spatial dose modulation and wet development. But there is also another, less obvious scenario of polymer reflow application in this case: e-beam exposure with dose modulation results in a nonhomogeneous distribution of the number average molecular weight ($M_{n}$) of the polymer layer after exposure. The regions with a lower $M_{n}$ will have a higher polymer chain mobility and, consequently, will tend to reflow faster than ones with a higher $M_{n}$. This effect could dramatically simplify the fabrication of complex three-dimensional structures—there would be no need for the formation of a stair-like relief provided that the greater part of the relief is formed in the reflow stage by a specially designed complex reflow process.

The problem is that there is no simulation method for the thermal reflow of polymeric structures obtained by GEBL. Existing methods based on transfer equations [3,10,11,12] require the homogeneous distribution of polymer viscosity, so this can not be applied for the simulation of nonhomogeneous polymeric structure reflow. On the other hand, there are numerical simulation methods that could potentially be applied in this case. For example, Kirchner and Schift [9] utilized numerical “soapfilm” modeling based on the finite element method for the thermal reflow simulation for a double-step structure obtained in PMMA by e-beam grayscale patterning and the subsequent wet development. The structure consisted of two regions with different $M_{n}$ values, which was taken into account by setting different vertex mobilities. The ratios of structure region mobilities were determined empirically by comparing the simulated profiles and the experimental ones. This method only allows thermal reflow simulations for the structures obtained with the same exposure doses, and reflow simulations for other structures require preliminary measurements. Kirchner [9] mentioned that the inverse of the mobility must be correlated with PMMA viscosity, but the relation between mobility and viscosity was still unclear, which means this method cannot be used in any other cases. Thus, the aim of this study was to develop the numerical approach to the thermal reflow simulation for any structure obtained in PMMA by grayscale e-beam lithography. For this purpose, one should first develop the method of viscosity profile determination for e-beam-exposed PMMA. Then, the relation between PMMA viscosity and the mobility of its surface vertices should be investigated.

## 2. Theory and Methods

Two methods of thermal reflow simulation for polymeric structures were used in this study. The first is the analytical reflow simulation method proposed by Leveder [11,13] for periodic structures obtained in polystyrene via nanoimprint lithography. This method is based on a two-dimensional Navier–Stokes equation coupled to a continuity equation considering Laplace pressure and Hamaker energy with the assumption of no slip length and no Marangoni effect. The initial structure profile undergoes Fourier transform and then the thermal reflow is simulated by the decay of the profile harmonic modes:(1)$$h\left(x,t\right)=h_{0}+\tilde{h}\left(x,t\right),$$
(2)$$\tilde{h}\left(x,t\right)=\sum_{-\infty }^{+\infty }a_{n}\left(0\right)\exp\left(-\frac{t}{\tau_{n}}+in\frac{2\pi }{\lambda }x\right),$$
(3)$$\tau_{n}=\frac{3\eta }{\gamma h_{0}^{3}}\times {\left(\frac{\lambda }{2\pi n}\right)}^{4},$$
where λ denotes the profile spatial periodicity, η and γ denote the polymer viscosity and surface tension, respectively, $a_{n}\left(0\right)$ denotes the Fourier coefficients of the initial profile, and $h_{0}$ denotes the polymer layer thickness. The polymer viscosity depends both on the temperature and polymer molecular weight, which should be taken into account. The temperature dependence of the viscosity could be described by the Williams–Landel–Ferry (WLF) equation [14]:(4)$$\log\left(\frac{\eta (T)}{\eta (T_{0})}\right)=-\frac{C_{1}\left(T-T_{0}\right)}{C_{2}+\left(T-T_{0}\right)},$$

The parameters $\eta (T_{0})$, $C_{1}$, $C_{2}$, and $T_{0}$ for the three different polymers are provided in Table 1.

**Table 1 polymers-15-03731-t001:** Parameters of Equation (Equation 4), obtained by Aho et al. [14] for polystyrene 143E by BASF (PS), polymethyl methacrylate Plexiglas 6N by Degussa (PMMA), and polycarbonate Lexan HF1110R by GE Plastics (PC).

Parameter	PS	PMMA	PC
$\eta (T_{0})$, Pa·s	7310.4	13,450	2763
$C_{1}$	10.768	7.6682	4.7501
$C_{2}$, °C	289.21	210.76	110.12
$T_{0}$, °C	190	200	200

The dependence of the viscosity on the polymer molecular weight could be described by the empirical formula:(5)$$\eta \propto M_{n}^{\alpha },$$
where $M_{n}$ denotes the number average polymer molecular weight. For polymethyl methacrylate (PMMA), α comprises 3.4 at $M_{n}>48,000$ and 1.4 at $M_{n}<48,000$ [11,15]. Equations (Equation 4) and (Equation 5) allow one to calculate the polymer viscosity for different temperatures and values of the number average molecular weight (Figure 1).

The second method, a numerical one, is based on a search of the minimal surface using the finite element method. It can be processed using the free software “Surface Evolver” (SE)—a program for the modeling of liquid surfaces shaped by various forces and constraints [16]. SE allows a wide spectrum of possible energies to be assigned, like gravitational energy and surface energy, and further different implementations of the mean and Gaussian curvature. In case of a thermal reflow simulation for polymeric structures obtained by lithographic methods, only the surface energy should be taken into account.

In the SE simulation algorithm, the structure is only described by its “outer shell” (soapfilm modeling). The structure surface is divided into triangle facets defined by vertices and oriented edges, $\vec{e_{0}}$, $\vec{e_{1}}$, and $\vec{e_{2}}$, and the polymer reflow is simulated by moving the facet vertices, and maintaining the constant volume inside the surface. Figure 2 shows an example of a triangulated surface model for an SE simulation, where $v_{0}$, $v_{1}$, $v_{2}$, and $\vec{e_{0}}$, $\vec{e_{1}}$, and $\vec{e_{2}}$ are the vertices and oriented edges of *i*-th facet, respectively.

The force acting on vertex $v_{0}$ is
(6)$${\vec{F}}_{v_{0}}=\frac{\gamma_{i}}{2}\cdot \frac{{\vec{e}}_{1}\times \left({\vec{e}}_{0}\times {\vec{e}}_{1}\right)}{∥{\vec{e}}_{0}\times {\vec{e}}_{1}∥},$$
where $\gamma_{i}$ denotes the surface tension of the *i*-th facet.

SE could be operated in the area normalization mode to approximate a vertex motion by mean curvature, i.e., a surface tension flow. In this mode, the velocity of a vertex is proportional to force acting on it and inversely proportional to the area of the facets surrounding this vertex. The *i*-th facet has three vertices associated with it; therefore, the relative area contribution to the force of one vertex is 1/3 the area of the surrounding facets *A*. The vertex velocity in the area normalization mode is
(7)$$\vec{v}=\frac{\vec{F}}{A/3}\cdot \mu ,$$
where μ is the so-called vertex mobility. The vector of the vertex movement $\vec{\delta }$ is then calculated as a product of the vertex velocity and the scale factor, which relates to the time of one iteration:(8)$$\vec{\delta }=\vec{v}\cdot scale.$$

In most cases, SE is used for the calculation of minimal energy geometries only [17,18,19], which does not imply the simulation of reflow dynamics. A significant reason for this usage scenario is the unclear relations between the SE simulation parameters (the mobility and total scale factor) and the physical ones (sample viscosity and reflow time).

## 3. Simulation of e-Beam-Exposed PMMA Viscosity Profile

In this study, a nonuniform profile of PMMA viscosity was supposed to arise from e-beam exposure at room temperature with dose modulation. According to Equations (Equation 4) and (Equation 5), one can calculate the PMMA viscosity for any specific temperature and number average molecular weight, so the number average molecular weight distribution of e-beam-exposed PMMA ($M_{n}$) is of interest.

Our approach to determine the $M_{n}$ distribution of the exposed PMMA bases in the simulation of e-beam-induced PMMA main-chain scissions is as follows. A Monte-Carlo simulation of e-beam scattering in the PMMA layer on the Si substrate was first carried out, taking into account the exposure parameters (electron beam energy, exposure dose and current, and spatial dose distribution). In the simulation algorithm, elastic scattering was modeled using Mott elastic scattering cross-sections obtained using the free software “ELSEPA” [20]; inelastic processes are described by models provided by Dapor [21] (for PMMA) and Valentin [22] (for Si). Next, according to Aktary [23], PMMA main-chain scissions were supposed to occur due to inelastic electron–electron scattering. Instead of using the common approach based on the PMMA radiation scission yield, electron–electron scattering events leading to PMMA main-chain scissions were simulated with the more accurate Monte-Carlo technique using the scission probability ($p_{s}$): (9)$$\mathrm{electron}-\mathrm{electron}\;\mathrm{scattering}:\left\{\begin{array}{cc} \xi <p_{s}: & \mathrm{scission}\\ \xi \geq p_{s}: & \mathrm{no}\;\mathrm{scission} \end{array}\right.$$
where ξ denotes a random number from the uniform distribution on [0, 1). The value of $p_{s}$ for room temperature (25 °C) was determined by the simulation of the experimental radiation scission yield using the approach described in our previous paper [24] and comprised 0.05.

Simulated PMMA main-chain scission events were stored in 3D histograms with a bin size of 50 nm. The example of the PMMA main-chain scission distribution simulated for the line exposure using this approach is shown in Figure 3.

Then, the number average molecular weight was calculated for each bin using the model of scissions randomly occurring at the bonds between monomers [25]:(10)$$\frac{1}{M_{n}^{\prime }}=\frac{w_{s}}{M_{0}}+\frac{1}{M_{n}},$$
where $M_{n}$ and $M_{n}^{\prime }$ denote the PMMA number average molecular weight before and after exposure, respectively, $M_{0}$ denotes the monomer molecular weight (100 for methyl methacrylate, MMA), and $w_{s}$ denotes the probability of scission at a bond. $w_{s}$ values were calculated for each bin using the formula:(11)$$w_{s}=\frac{N_{sci}}{N_{mon}},$$
where $N_{sci}$ denotes the number of scissions and $N_{mon}$ the number of monomers in the bin. The number of monomers in the (50 nm)${}^{3}$ bin was calculated from the PMMA density (1.19 g/cm${}^{3}$) and MMA molecular weight (100 g/mol) and comprised 894,809. The example of the $M_{n}^{\prime }$ distribution simulated for the line exposure by this method is shown in Figure 4.

The obtained $M_{n}^{\prime }$ distribution of the exposed PMMA was highly informative for two reasons. Firstly, it could be directly converted to the distribution of the PMMA solubility rate using the empirical Greeneich [26] formula:(12)$$R=R_{0}+\frac{\beta }{{\left(M_{n}^{\prime }\right)}^{\alpha }},$$
where parameters $R_{0}$, α, and β depend on the solvent and could be found in Greeneich’s papers [26,27]. The simulation technique to resist wet development in the case of a known resist solubility rate distribution is common and its description could be found elsewhere [28,29].

Secondly, the exposed PMMA $M_{n}^{\prime }$ distribution could be used for the simulation of the PMMA viscosity distribution for the required temperature using Equations (Equation 4) and (Equation 5). This is the key parameter in our study and, for the following simulation, the viscosity distribution was averaged along the *z* axis, which resulted in a viscosity profile of the e-beam-exposed PMMA (Figure 5).

## 4. Determination of e-Beam-Exposed PMMA Vertex Mobilities

The obtained PMMA viscosity profile could not be used for the thermal reflow simulation—the analytical approach based on the profile Fourier transform could only be applied in case of uniform viscosity. On the other hand, the numerical approach allows reflow simulations for nonuniform structures, but it is based on vertex mobilities not viscosity. Therefore, one should investigate the relation between the polymer viscosity and vertex mobilities of its surface. For this purpose, thermal reflow was simulated by both approaches for PMMA rectangular gratings, the parameters of which corresponded to Leveder’s study [11]—a 2 μm pitch and 28 nm depth. For PMMA viscosity values in the range 10${}^{2}$–10${}^{6}$ Pa·s, grating reflow was simulated analytically with constant time steps. Then, the initial grating surface was reconstructed in SE and the surface evolution during grating reflow was simulated numerically with vertex mobilities equal to 1. During the numerical simulation, total scale values, which gave the same grating profiles as ones obtained using analytical approach, were determined (Figure 6). It was found that, in the beginning of reflow, there is a slight discrepancy between profiles simulated analytically and numerically, but then both approaches lead to an sinusoidal almost shape as is predicted by Equations (Equation 1)–(Equation 3). Time-scale data obtained for different viscosity values showed an almost direct proportionality between the total scale and time (*t*) (Figure 7):(13)scale≈α·t.

The values of α obtained by the approximation of the *t*-scale data using Function (Equation 13) demonstrated quite a linear dependence of ln(α) on ln(η) (Figure 8):(14)ln(α)≈a·ln(η)+b.

The approximation of η–α data by the function
(15)$$\alpha =\frac{C}{\eta^{\beta }}$$
resulted in *C* and β values equal to 26.142 and 0.989. Thus, there is almost inverse proportionality between α and PMMA viscosity:(16)$$\alpha \approx \frac{26.142}{\eta }.$$

The determined relation between PMMA viscosity and α (which represents the ratio of the total scale to *t*) enables mobility-based thermal reflow simulations using SE. The point is that SE allows the total scale value to be monitored during the simulation; however, originally, the total scale is not equal to the reflow time. The most convenient relation between the total scale and time would be an equality (total scale≡t) and it could be achieved by setting the vertex mobility equal to α. Indeed, if one simultaneously set
(17)$$\left\{\begin{array}{c} \mu \equiv \alpha =\frac{scale}{t},\\ scale=t, \end{array}\right.$$
Equation (Equation 7) does not change:(18)$$\vec{\delta }=\frac{\vec{F}}{A/3}\cdot \frac{scale}{t}\cdot t\equiv \frac{\vec{F}}{A/3}\cdot scale.$$

Equation (Equation 17) is the final piece in the mobility-based thermal reflow simulation for e-beam-exposed PMMA. The viscosity profile of PMMA, obtained in a previous step, could be easily turned into a mobility profile using the following formula (Figure 9):(19)$$\mu =\frac{26.142}{\eta },$$
where η is in Pa·s.

Thus, one can reconstruct in SE the surface of the structure obtained in PMMA after wet development, set proper mobilities of the surface vertices, and then run an SE simulation tracking the total scale factor, which will be exactly equal to the reflow time.

## 5. Discussion

The developed approach to the simulation of the thermal reflow of PMMA structures has several remarkable features. Firstly, it demonstrates the agreement between reflowed profiles simulated by an analytical spectral method and those obtained by a numerical SE simulation in area normalization mode. This emphasizes the applicability of soapfilm modeling for the simulation of polymer thermal reflow and confirms the linear relation between the total scale factor and time, proposed by Kirchner [9]. Moreover, the relation of the inverse mobility of PMMA surface vertices and PMMA viscosity turns out to also be linear. It is also noteworthy that the developed method brings clarity into the SE reflow simulation process—in the case of setting the mobilities obtained by Equation (Equation 19), the total scale factor exactly denotes the actual reflow time.

Secondly, this method allows a thermal reflow simulation for any structure obtained in PMMA using lithographic methods. In the case profile obtained in PMMA by grayscale e-beam lithography, the simulation of the PMMA main-chain scission distribution could be directly converted to a PMMA viscosity profile and, finally, to a PMMA vertex mobility profile. In the case of structures obtained by nanoimprint lithography, only the PMMA viscosity dependence on temperature and molecular weight should be taken into account. If sample size is much greater than the structure spatial periodicity, one can neglect the edge effects (contact angle, etc.) for the reflow simulation far from the sample edges. Then, the structure geometry and vertex mobility distribution become the only parameters required for the simulation.

Thirdly, the described algorithm expands the boundaries of the thermal reflow application. At present, thermal reflow is used predominantly for profile smoothing, which does not cause a dramatic profile transformation [8]. This results in a need for a sophisticated grayscale e-beam lithography process to achieve a staircase profile close enough to the required one. The complex but predictable thermal reflow stage could, in turn, simplify the whole fabrication process provided that a more significant part of the profile is being formed by reflow.

## 6. Conclusions

This paper presents a new simulation method for the thermal reflow of PMMA with a nonuniform viscosity profile caused by e-beam exposure with dose modulation. This method includes three steps:The Monte-Carlo simulation of e-beam-induced PMMA main-chain scissions for the determination of the PMMA viscosity profile;The calculation of the mobilities of the PMMA surface vertices;The PMMA reflow simulation, which is processed by a numerical search of the minimal surface.

The relation between PMMA viscosity and the vertex mobility of its surface was determined by simulating the rectangular grating reflow using two approaches—an analytical approach based on the decay simulation of the profile spatial harmonics, and a numerical approach based on a finite element method. Using this relation, one can process a comprehensible reflow simulation for a nonhomogeneous PMMA structure using the free software “Surface Evolver”. The developed method provides deeper insights into the thermal reflow of polymers, which could simplify the fabrication of complex three-dimensional structures.

## Figures and Tables

**Figure 1 polymers-15-03731-f001:**
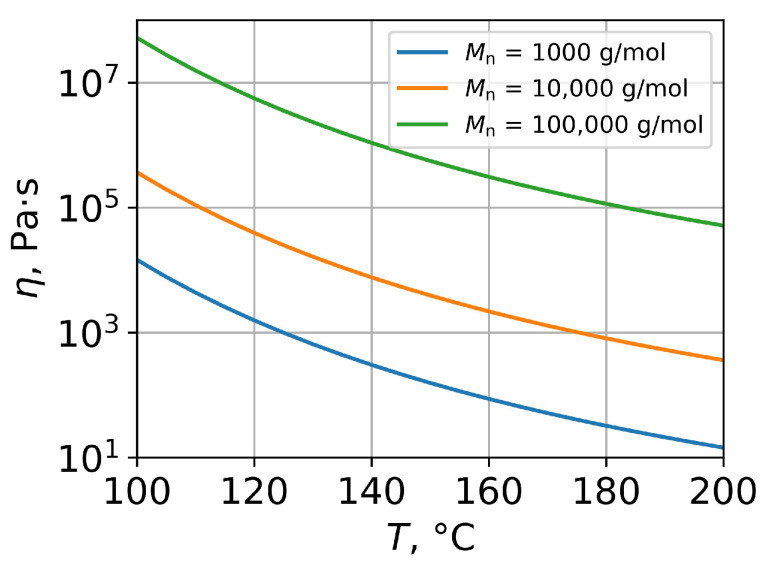
Temperature viscosity dependencies for PMMA with different number average molecular weights, obtained by Equations (Equation 4) and (Equation 5).

**Figure 2 polymers-15-03731-f002:**
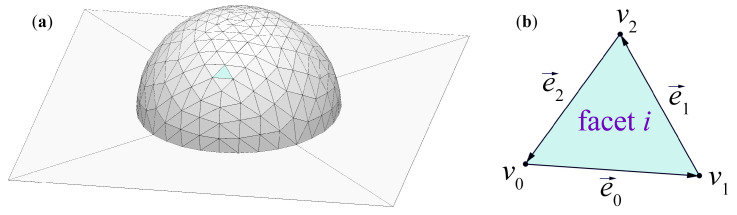
(**a**) A mound of liquid sitting on a tabletop and defined by its surface in SE simulation. (**b**) Definition of vertices and oriented edges of *i*-th facet in SE.

**Figure 3 polymers-15-03731-f003:**
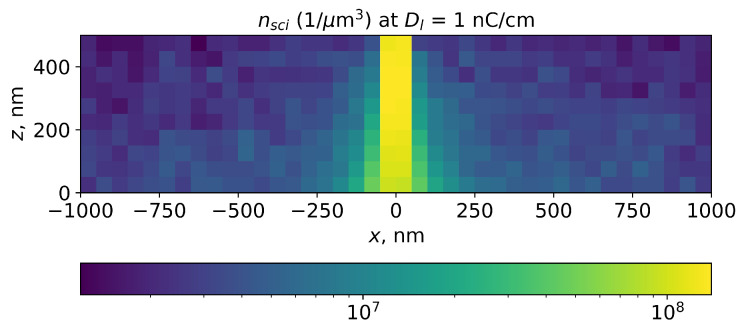
Simulation of local PMMA main-chain scission concentration in PMMA layer for line exposure. Line dose is 1 nC/cm, e-beam energy is 20 keV, and PMMA layer thickness is 500 nm. Scission probability in inelastic electron–electron scattering is 0.05, which corresponds to room temperature (25 °C).

**Figure 4 polymers-15-03731-f004:**
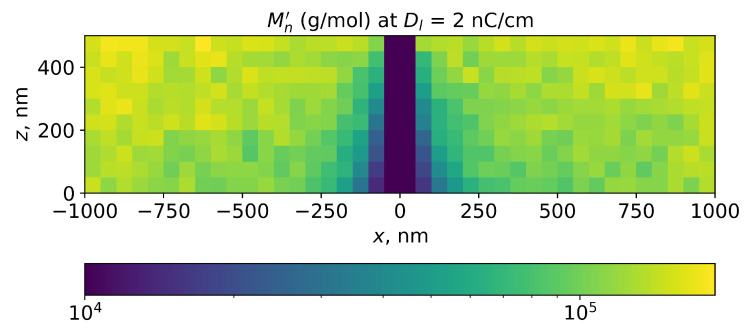
Simulation of local number average PMMA molecular weight in PMMA layer for line exposure at room temperature. Line dose is 1 nC/cm, e-beam energy is 20 keV, and PMMA layer thickness is 500 nm. The initial PMMA number average molecular weight is 271,000.

**Figure 5 polymers-15-03731-f005:**
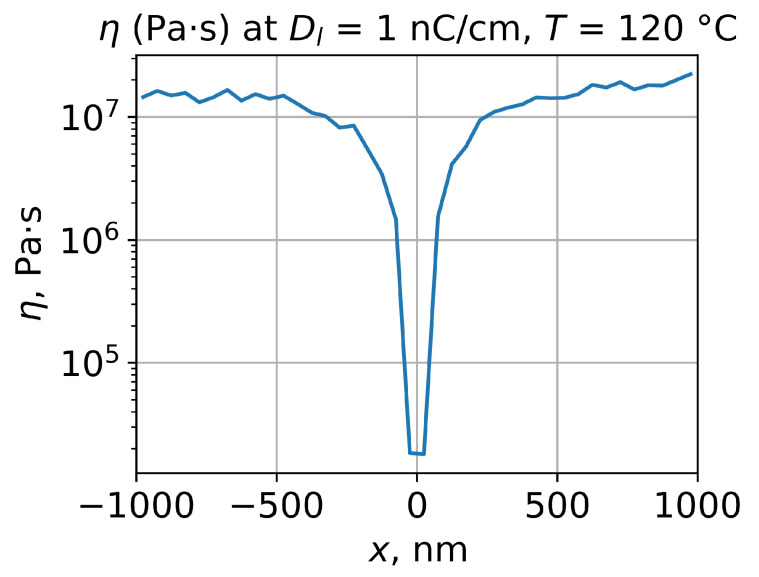
Simulation of viscosity profile of e-beam-exposed PMMA layer corresponding to the temperature 120 °C. Line dose is 1 nC/cm, e-beam energy is 20 keV, and PMMA layer thickness is 500 nm. The initial PMMA number average molecular weight is 271,000.

**Figure 6 polymers-15-03731-f006:**
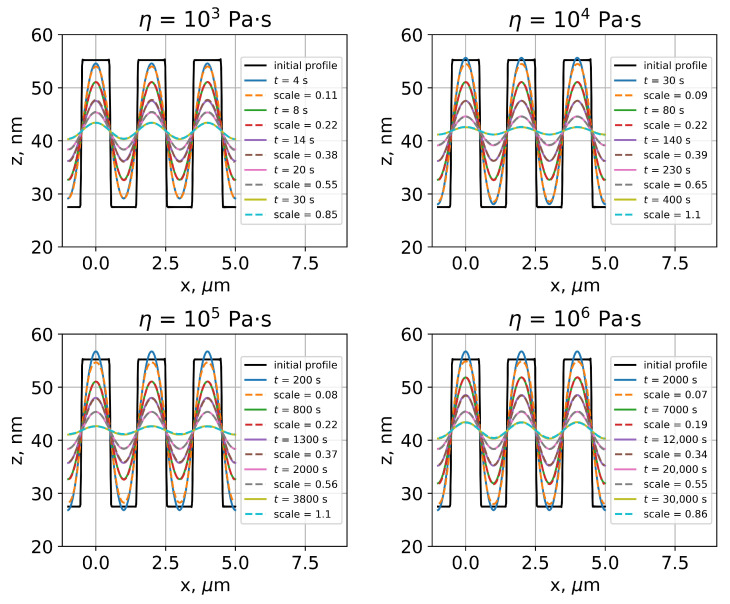
Thermal reflow of PMMA rectangular grating simulated by analytical and numerical approaches for different PMMA viscosity values. For clarity, only three grating periods are shown.

**Figure 7 polymers-15-03731-f007:**
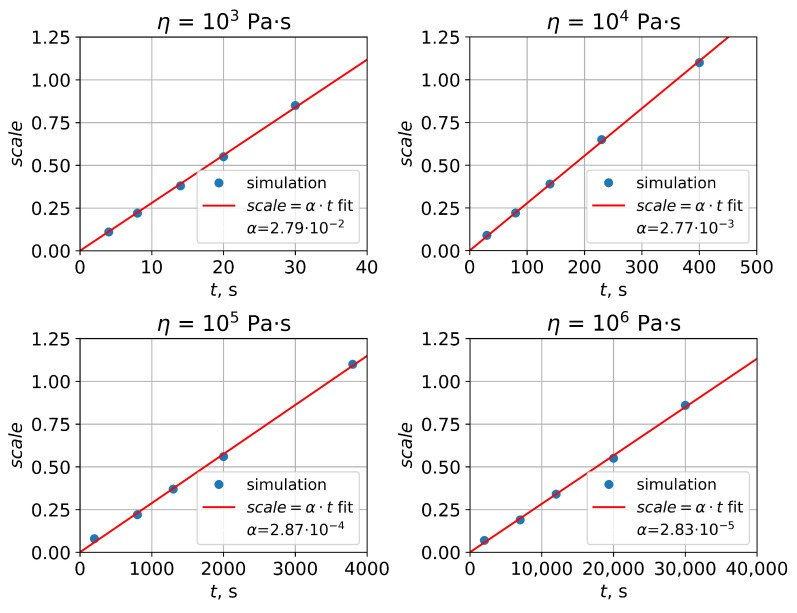
Time-scale dependencies for different viscosity values fitted with linear functions.

**Figure 8 polymers-15-03731-f008:**
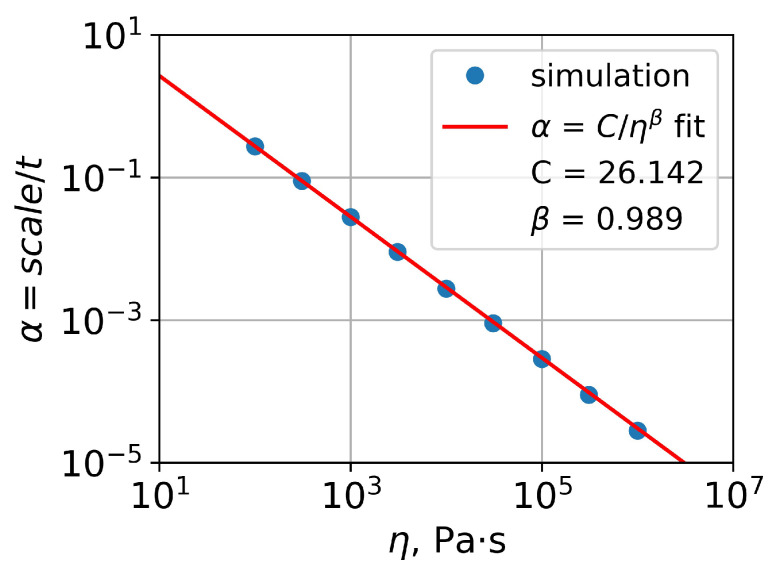
Fit of obtained η-α data for PMMA gratings.

**Figure 9 polymers-15-03731-f009:**
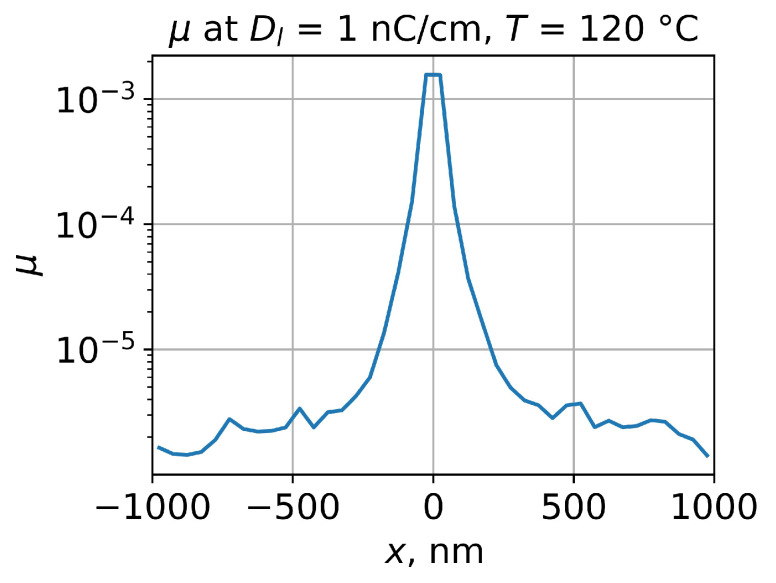
Simulation of averaged (along *z* axis) vertex mobility profile of e-beam-exposed PMMA layer for 120 °C. Line dose is 1 nC/cm, e-beam energy is 20 keV, PMMA layer thickness is 500 nm. The initial PMMA number average molecular weight is 271,000.

## Data Availability

The data used to support the findings of this study are included within the article.

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
