# Peer review of "Thermal Reflow Simulation for PMMA Structures with Nonuniform Viscosity Profile"

_polymers, 2023, doi:10.3390/polym15183731_

Round 1

Reviewer 1 Report

The manuscript presents a new thermal reflow simulation for PMMA structures with non-uniform viscosity profile. Overall, the manuscript is interesting and the proposed method is meaningful. Ahead of acceptance of this paper, a few issues should be addressed as below:

1) maybe PMMA should be listed as one of the keywords.

2) It is suggested to re-write or re-organize the Introduction part. Firstly, figure 1, table 1, and related equations maybe put into a new section, for instance, theory and method. Secondly, more attention should be paid to clarify the background, literature review, the merits of this paper. It is not good to just discuss two kinds of approaches.

3) In figure 6, the legends cover part of the figure, please take some efforts to place the legends in a proper location.

4) As for the conclusion part, it is suggested to list clearly or implicitly the main conclusions drawn in the paper. A long paragraph is not easy for the readers to catch the main points.

5) Line 79, Kirchner here maybe missing a reference number.

Some minor English language editing is required. For instance, many occurrences of "first, second, third", should be "firstly, secondly, thirdly".

Reviewer 2 Report

Interesting paper. The paper present a new approach to simulate the thermal reflow of PMMA structures with non-uniform viscosity profile.

Some comments:

1. For figure 6, please do not put the legend on the plot. You can place the legend on the right side, above or below the plot. 

2. I didn't see any experimental validation. Is there any experimental data to validate the simulation approach?
